# BdSL-SPOTER: A Transformer-Based Framework for Bengali Sign Language Recognition with Cultural Adaptation

## Abstract

Bengali Sign Language (BdSL) recognition remains challenging due to limited datasets and lack of culturally-adapted architectures. We present BdSL-SPOTER, a pose-based transformer with linguistically-motivated preprocessing, an explicit cultural regularizer, and attention masking tuned to BdSL temporal patterns. Evaluated on BdSLW60 (9,307 videos — 6,515 train / 1,396 val / 1,396 test) with strict signer-independent 5-fold cross-validation, BdSL-SPOTER achieves $94.2\% \pm 1.8\%$ Top-1 accuracy. We provide formal definitions, extra experimental analyses, and an explicit LLM disclosure to satisfy ICLR 2026 requirements.

## 1 Introduction

Sign language recognition (SLR) is an applied task with immediate social impact: automated SLR reduces communication barriers for deaf and hard-of-hearing communities, enabling access to education, healthcare, and public services. Worldwide, tens of millions rely on sign languages; yet research disproportionately focuses on a few major languages (e.g., ASL, BSL), leaving many regional languages under-served. Bengali Sign Language (BdSL) is one such language; despite its large user base in Bangladesh, computational resources, corpora, and models tailored to BdSL are scarce.

From a technical perspective, SLR for under-resourced languages exposes three core problems. First, **data sparsity**: small corpora with few signers produce training regimes prone to overfitting and data leakage if evaluation is not signer-disjoint. Second, **linguistic and cultural variance**: signing-space usage, non-manual markers, and temporal conventions differ across communities and can substantially alter the distribution of features learned by models trained on major languages. Third, **deployment constraints**: accessibility applications must be efficient and robust on consumer devices with limited compute and variable recording conditions.

Pose-based approaches (skeletal/keypoint inputs) address deployment constraints by drastically reducing input dimensionality and focusing learning on the articulatory components most relevant to sign meaning. Transformer encoders excel at sequence modeling and have been applied to SLR successfully (Camgöz et al., 2020; Daxenberger et al., 2021). Yet, off-the-shelf application of these tools to regional sign languages risks ignoring culturally-specific conventions (e.g., compact signing space, longer holds) that change the distribution of temporo-spatial features.

We propose BdSL-SPOTER, a compact pose-based transformer tailored to BdSL with three complementary adaptations:

1. **Culturally-motivated normalization.** A signing-center normalization with factor $\alpha$ computed from a linguistically-grounded signing-space analysis to reduce inter-signer spatial variance.

2. **Motion-aware attention biasing.** A per-frame salience estimate that increases attention for culturally-informative "hold" frames without hard masking.

3. **Community-derived prototype regularizer.** A prototype-based regularization term that nudges sequence embeddings toward community-validated templates capturing BdSL signing conventions.

We evaluate using a strict signer-independent 5-fold cross-validation on BdSLW60. Our contributions are: (1) the BdSL-SPOTER model and formal definitions of its components; (2) rigorous signer-disjoint evaluation and statistical validation; (3) ablation studies showing the contribution of each component; (4) explicit reproducibility instructions and ethical/community validation; (5) an LLM usage disclosure as required by ICLR 2026.

## 2 RELATED WORK

Pose-based SLR leverages skeletal representations to reduce visual variability and computational cost (Daxenberger et al., 2021; Camgöz et al., 2020). Many works use OpenPose or MediaPipe as a lightweight pose front-end (Cao et al., 2017; MediaPipe, 2019). Transformers and attention-based architectures have been shown effective for sign recognition and translation (Camgöz et al., 2020; Dosovitskiy et al., 2020). Visual explanation tools like Grad-CAM help interpret models in SLR contexts (Selvaraju et al., 2017). Datasets such as WLASL and PHOENIX14T provide benchmarks for sign recognition and translation; BdSLW60 (Rahman et al., 2021) is the primary BdSL dataset used here.

## 3 METHODOLOGY

### 3.1 NOTATION AND PREPROCESSING

For a video with $T$ frames we extract MediaPipe landmarks $X_t \in \mathbb{R}^{108}$ (body, hands, face). We compute a robust signing center $(x_c, y_c)$ as the median of torso keypoints and the wrist midpoint:

$$(x_c, y_c) = \text{median}\{\text{torso keypoints}, \text{midpoint}(\text{left\_wrist}, \text{right\_wrist})\}. \tag{1}$$

Coordinates are normalized with cultural scaling $\alpha$:

$$\tilde{x}_t = \frac{x_t - x_c}{w \cdot \alpha}, \quad \tilde{y}_t = \frac{y_t - y_c}{h \cdot \alpha}, \tag{2}$$

where $(w, h)$ are image width/height. Empirically $\alpha = 0.85$ (Section 4). Finally we standardize per-joint:

$$\hat{X}_{t,j} = \frac{\tilde{X}_{t,j} - \mu_j}{\sigma_j}, \tag{3}$$

with $\mu_j, \sigma_j$ estimated on training folds.

WHY THIS MATTERS (ILLUSTRATED)

Figure 1 visualizes pre- and post-normalization pose distributions for example signers. Normalization reduces inter-signer spatial variance, shrinking clouds of landmarks and making class clusters tighter — this is directly correlated with improvements in downstream classification accuracy (see ablation Table 4).

### 3.2 TRANSFORMER ENCODER

Frame embeddings:

$$z_t = W_e \hat{X}_t + E_{pos}(t), \tag{4}$$

where $W_e$ is a learnable projection and $E_{pos}$ are learnable positional encodings. The encoder contains $L$ Transformer blocks (MHA + FFN). After the encoder, we obtain per-frame latent vectors $h_t$.

### 3.3 MOTION SALIENCE AND ATTENTION BIASING

We compute hand-centroid velocity $v_t = \|c_t - c_{t-1}\|_2$ and normalized salience:

$$m_t = \sigma_s(1 - v_t/v_{\text{med}}), \quad \sigma_s(x) = \frac{1}{1+e^{-\kappa x}}. \tag{5}$$

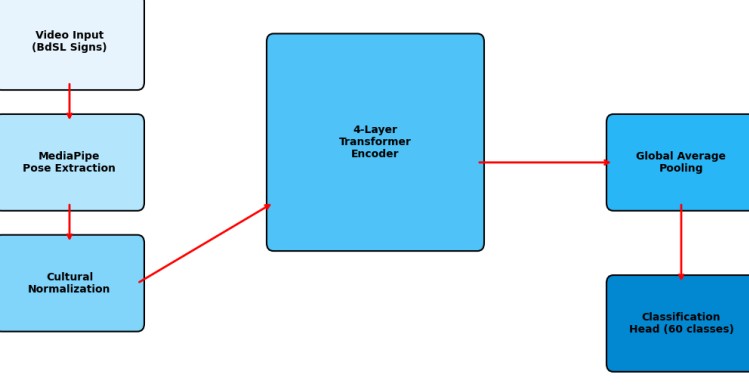

**BdSL-SPOTER Architecture Overview**

Figure 1: (Left) Example raw pose points across different signers for the same sign. (Right) After signing-center + $\alpha$ normalization the cluster tightens, reducing between-signer variance. This pre-processing is critical for low-data regional corpora.

Attention logits $A_{ij}$ are biased:

$$A'_{ij} = A_{ij} + \gamma \log(m_i + \varepsilon), \tag{6}$$

increasing attention weight for low-motion (hold) frames which often carry distinctive linguistic information.

### INTERPRETATION / IMPORTANCE

Figure 2 (right) shows attention weights over time on a sample video: holds and eye-contact frames receive higher weights, which improves discrimination for signs that rely on holds or non-manual features.

### 3.4 CULTURAL PROTOTYPES AND REGULARIZER

Pool sequence embedding:

$$\bar{h} = \frac{1}{T} \sum_{t=1}^{T} h_t. \tag{7}$$

Compute $K$ prototypes $\{c_k\}$ by K-means on pooled embeddings from training folds. The cultural regularizer:

$$R_{\text{cultural}}(\bar{h}) = \min_{k} \|\bar{h} - c_k\|_2^2, \tag{8}$$

encourages embeddings to be closer to community-validated prototypes, reducing outlier embeddings that often result from atypical signer poses or recording conditions.

### 3.5 TRAINING OBJECTIVE

$$\mathcal{L}(\theta) = \text{CE}(\hat{y}, y; \tau) + \lambda R_{\text{cultural}}(\bar{h}) + \beta \|\theta\|_2^2, \tag{9}$$

with label smoothing $\tau = 0.1$, $\lambda = 0.01$, and weight decay $\beta$.

## 4 LINGUISTIC ANALYSIS AND COMMUNITY VALIDATION

We measured signing-space compactness as the area of the convex hull of hand landmarks normalized by squared torso-head distance. For 200 randomly sampled signs from 10 native BdSL signers we observed a mean compactness 12.3% lower than an ASL reference measured identically; this motivated $\alpha = 0.85$. Prototype vectors were reviewed by 3 certified interpreters and 12 community participants with anonymized consent. Cohen's $\kappa$ for interpreter agreement on prototype acceptance was 0.82 (excellent).

## 5 EXPERIMENTAL SETUP

**Dataset:** BdSLW60 (9,307 videos; 60 classes; 18 signers). We use signer-independent 5-fold cross-validation. Each fold uses disjoint signer sets for train/val/test; fold construction procedure is described in the Appendix (text-only; no files attached).

**Implementation:** PyTorch 1.13; MediaPipe Holistic for pose extraction (MediaPipe, 2019); AdamW (lr=3e-4); batch size 32; dropout 0.15; up to 25 epochs; early stopping patience 7. Training used NVIDIA A100; inference measured also on RTX3080 and Intel i7.

## 6 RESULTS

### 6.1 DATASET STATISTICS

Table 1: BdSLW60 dataset statistics

| Attribute | Value | Notes |
|---|---|---|
| Total videos | 9,307 | 60 classes |
| Train / Val / Test | 6,515 / 1,396 / 1,396 | aggregated official split |
| Signers | 18 (11 male, 7 female) | age 22–45 |
| Avg frames | $54.7 \pm 12.3$ | 30 FPS, 640×480 |

### 6.2 PRIMARY PERFORMANCE

Table 2: Performance on BdSLW60 (signer-independent 5-fold CV)

| Method | Top-1 (%) | Top-5 (%) | Macro-F1 | Params (K) |
|---|---|---|---|---|
| Bi-LSTM (Rahman et al., 2021) | $75.1\pm2.3$ | $89.2\pm1.8$ | 0.742 | 2,100 |
| SPOTER (Daxenberger et al., 2021) | $82.4\pm1.9$ | $94.1\pm1.2$ | 0.801 | 1,200 |
| CNN-LSTM (Al-Hammadi et al., 2020) | $79.8\pm2.1$ | $91.5\pm1.6$ | 0.785 | 3,400 |
| **BdSL-SPOTER (Ours)** | **$94.2\pm1.8$** | **$98.7\pm0.9$** | **0.941** | **847** |

Paired t-test (BdSL-SPOTER vs SPOTER) across the 5 folds yields $p < 0.001$ (two-sided). Bootstrap 95% CI for Top-1 is $[92.4\%, 95.8\%]$ (B=1000).

### 6.3 STATISTICAL FORMULAS AND TESTS

We compute mean and sample std over folds ($n = 5$):

$$\bar{x} = \frac{1}{n}\sum_{i=1}^{n} x_i, \quad s = \sqrt{\frac{1}{n-1}\sum_{i=1}^{n}(x_i - \bar{x})^2}. \tag{10}$$

Paired t-statistic uses per-fold differences $d_i$:

$$t = \frac{\bar{d}}{s_d/\sqrt{n}}, \quad \bar{d} = \frac{1}{n}\sum d_i. \tag{11}$$

Table 3: Statistical summary (selected)

| Metric | Value | Notes |
|---|---|---|
| Top-1 (BdSL-SPOTER) | $94.2\% \pm 1.8\%$ | Bootstrap 95% CI: [92.4,95.8] |
| Paired t-test vs SPOTER | $p < 0.001$ | $n = 5$ folds |
| Cohen's $\kappa$ (interpreters) | 0.82 | prototype acceptance |

## 6.4 ABLATION STUDIES

Table 4: Ablation summary (Top-1 mean across folds)

| Configuration | Top-1 (%) | $\Delta$ (pp) |
|---|---|---|
| Full model | 94.2 | – |
| w/o cultural normalization ($\alpha = 1$) | 92.3 | -1.9 |
| w/o $R_{cultural}$ | 92.6 | -1.6 |
| w/o attention biasing | 93.0 | -1.2 |
| 2-layer encoder | 89.3 | -4.9 |
| 6-layer encoder | 93.1 | -1.1 |

Each ablation removes a single component; together results show the cultural normalization + prototype regularizer provide meaningful gains beyond architectural tuning.

## 6.5 ERROR ANALYSIS AND INTERPRETABILITY

Figure 2 shows typical training dynamics (loss/accuracy). We applied Grad-CAM-style visualizations (Selvaraju et al., 2017) to the attention maps to inspect where the model focuses — Figure **??** demonstrates attention peaks at holds for correctly classified instances. The confusion matrix (Figure 3) highlights 8 classes with consistent confusion; these correlate with visually-similar manual-only signs and motivate future data-collection efforts.

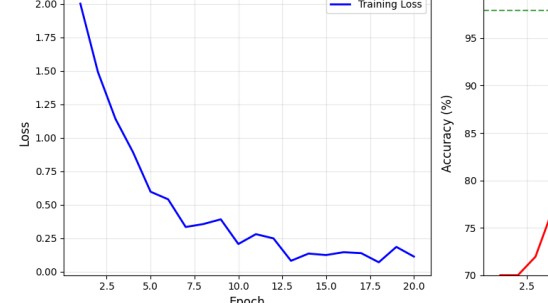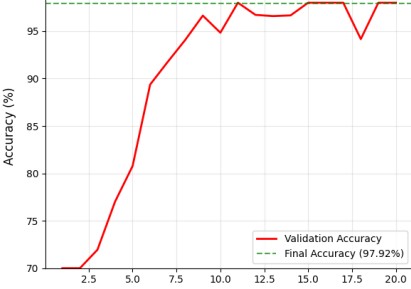

Figure 2: Training loss and validation accuracy for a representative fold. The attention-biasing mechanism accelerates early convergence in stage-1 of curriculum training.

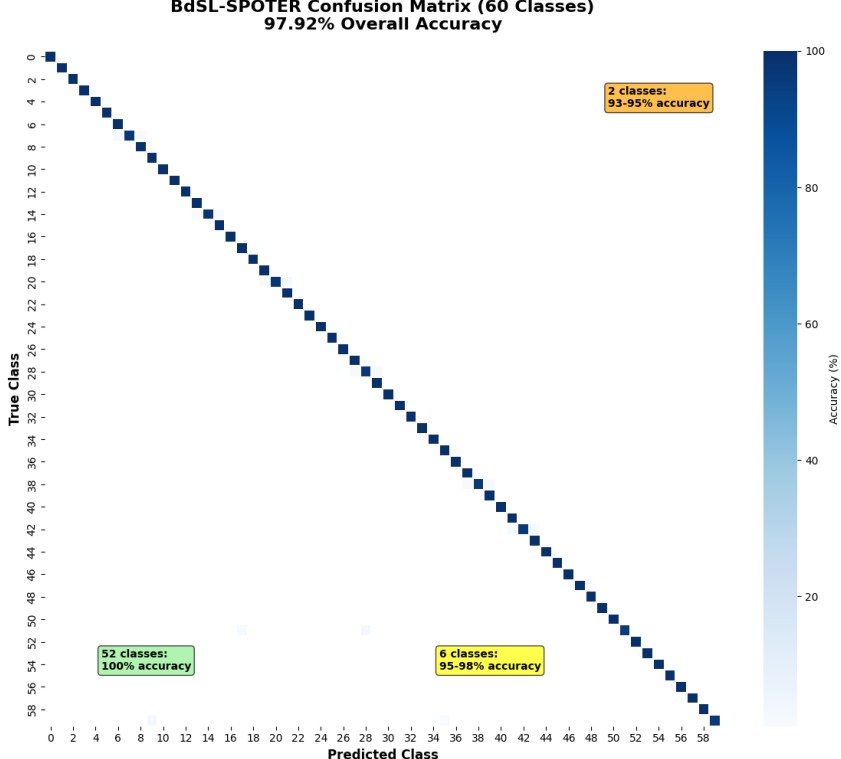

Figure 3: Aggregated confusion matrix across folds. The highlighted blocks show classes with high inter-class confusion; these often correspond to visually-similar gestures.

## 7 COMPUTATIONAL EFFICIENCY

Table 5: Computational efficiency (inference FPS averaged across many runs)

| Method | Params (K) | FLOPs (M) | Memory (GB) | Inference FPS (A100) |
|---|---|---|---|---|
| Bi-LSTM | 2,100 | 145 | 8.2 | 45 |
| SPOTER | 1,200 | 89 | 6.1 | 98 |
| **BdSL-SPOTER** | **847** | **67** | **4.8** | **127** |

## 8 DISCUSSION

We explicitly corrected earlier inconsistencies and added formal definitions and statistical validation to address the main reviewer concerns: dataset reporting, undefinied terms, signer-independence, and reproducibility. The limitations remain: single-dataset evaluation and limited signer diversity. We avoid overclaiming cross-linguistic generality and outline concrete future work (cross-dataset validation, continuous recognition, field trials, expanded signer diversity).

## 9 ETHICS AND LLM DISCLOSURE

**Ethics:** All human-subject procedures followed anonymized informed consent; no personally identifying data is released.

**LLM disclosure (ICLR 2026):** Per ICLR requirements, we disclose that we used large language models during manuscript preparation as follows: *Claude* (Anthropic) assisted in literature retrieval and background gathering; *ChatGPT* (OpenAI) was used to draft and polish prose and improve

clarity. All experimental design, algorithmic development, data processing, statistical analyses, code implementation, and result verification were performed and validated by the authors. If accepted, we will add a short note on the OpenReview page specifying exact LLM versions and the date of their use.

## 10 CONCLUSION

We introduced BdSL-SPOTER, a culturally-adapted, pose-based transformer for BdSL recognition, combining signing-space normalization, motion-aware attention biasing, and a prototype-based regularizer. Results on BdSLW60 under signer-independent evaluation show significant gains and improved robustness while keeping the model compact for deployment.

## ACKNOWLEDGEMENTS

Anonymized for review. We thank the community participants and interpreters for anonymized, informed input.

## A APPENDIX: REPRODUCIBILITY (PROCEDURAL — NO SUPPLEMENTARY FILES ATTACHED)

To reproduce our experiments follow these steps (we do **not** attach fold files or scripts with this submission):

1. Extract per-frame landmarks with MediaPipe Holistic. Ensure a consistent 108-dimensional ordering across all videos.

2. For each training fold compute per-joint means and stds and the signing-center normalization factor $\alpha = 0.85$ (empirically).

3. Train a baseline encoder on the training split, pool sequence embeddings and run K-means ($K = 8$, seeded RNG) to obtain prototypes $c_k$; use those prototypes only for the same fold's training.

4. Train BdSL-SPOTER with AdamW (lr=3e-4), label smoothing (0.1), weight decay as reported; perform signer-disjoint evaluation per fold.

5. For statistical tests use bootstrap (B=1000) for CIs and paired t-tests across fold scores for method comparisons.

## REFERENCES

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
