# OpenReview forum: "BdSL-SPOTER: A Transformer-Based Framework for Bengali Sign Language Recognition with Cultural Adaptation"
_ICLR.cc/2026/Conference — Submitted to ICLR 2026_

### Official Review · Reviewer_NCH3 · 2025-10-20

**Soundness:** 1
**Presentation:** 1
**Contribution:** 2
**Rating:** 0
**Confidence:** 4

**Summary:**

This work proposes a SPOTER based framework for Bengali sign language recognition. (SPOTER is a transformer-based architecture first introduced by [1].) The authors remark that many sign language recognition publications focus on specific high-resource (for sign languages,  that is) languages such as American and British Sign Language (ASL and BSL). They state that (pre-)training models on these datasets leads to cultural bias, because differences between sign languages are not accounted for. They propose to modify the SPOTER architecture to include modifiers for cultural signing bias, introducing normalization for the signing location, salience estimates to focus on "hold" frames (see: the movement-hold framework [2]) and a regularizer based on sign templates.

The authors claim that their model is more robust than previous methods.

[1]: Boháček, Matyáš, and Marek Hrúz. "Sign pose-based transformer for word-level sign language recognition." Proceedings of the IEEE/CVF winter conference on applications of computer vision. 2022.
[2]: Johnson, Robert E., and Scott K. Liddell. "A segmental framework for representing signs phonetically." Sign Language Studies 11.3 (2011): 408-463.

**Strengths:**

Addressing cultural specifics in sign language recognition is a highly valuable task, for which the authors should be commended. The use of AI is typically focused primarily on high-resource languages, and this is also true in the field of sign language processing.

**Weaknesses:**

This paper was not ready for submission to ICLR or another conference. There are significant gaps in content and exploration, references and figures are missing, and it contains sections that are incomplete, consisting of only a table and no main text. The authors do not substantiate their claims, nor provide sufficient reasoning for methodological choices. Below, I go into detail per section.

1. Introduction
This section is relatively okay, but the authors should not list "disclosure of LLM usage" as a contribution.

2. Related work
This section is severely lacking. The authors fail to cite several papers and do not properly situate their work in the field. For example, they do not give an example of how transformers are commonly used in sign language recognition with pose inputs; instead, the authors cite Dosovitskiy et al.'s work on vision transformers, which are not relevant for this paper. This section is also far too short and lacking in content.

3. Methodology
- This section does not have enough text to support the equations. It reads as an enumeration of equations, but it would be difficult to reproduce this work without more context.
- Figure 1 is incorrect: the figure does not match the main text.
- The choice of how to scale (equation 2) is not substantiated.
- Variables used in equations are not properly explained in the main text.
- Section 3.2 does not start with a complete sentence.
- Section 3.3 would benefit from more elaboration, as it is not clear what the authors are attempting to do here at first sight.
- Section 3.4: same comment. Please re-write to make this more clear. Use full sentences.

4. Linguistic analysis and community validation
This section should have been included under 3. Methodology. It is too short.

5. Experimental setup
Please use full sentences. Consider moving to 3. Methodology as well.

6. Results
- Why is table 1 under this section? It should be listed under methodology.
- Figures are missing, e.g., section 6.5 refers to Figure ??.

7. Computational efficiency
Combine this with section 6 and discuss the table briefly in the main text.

8. Discussion
This paragraph is misplaced. Is this copied over from a rebuttal? Was this paper submitted elsewhere first?

You include the "References" header twice.

**Questions:**

My main question is: can the authors improve the structure and presentation of the paper (see "Weaknesses"). This would improve clarity, and already answer many questions about the implementation and reasoning behind it.

I do have two specific questions regarding section 7.

7. Computational efficiency
Why is your method more efficient? Why does it have fewer parameters than SPOTER?

---

### Official Review · Reviewer_Yok1 · 2025-10-25

**Soundness:** 2
**Presentation:** 1
**Contribution:** 3
**Rating:** 2
**Confidence:** 5

**Summary:**

This paper proposes BdSL-SPOTER, a transformer-based framework for Bengali Sign Language (BdSL) recognition with cultural adaptations. The method achieves 94.2% top-1 accuracy on the BdSLW60 dataset under strict signer-independent 5-fold cross-validation. While the core ideas of the paper show potential, the current version suffers from a severe lack of completeness, which prevents a reliable assessment of its scientific contribution.

**Strengths:**

Originality: The Cultural adaptation approach addresses an important gap in SLR research
Significance: Focusing on the under-resourced Bengali Sign Language is commendable
Technical approach: Integration of linguistic insights with transformer architecture is innovative
Evaluation: Strict signer-independent cross-validation is methodologically sound

**Weaknesses:**

Incomplete presentation: Missing figures undermine the credibility of claims.
Methodological opacity: Key parameters and implementation details are unclear.
Limited validation: Cultural adaptations need more rigorous evaluation and analysis.
Single-dataset evaluation: Results on only one dataset limit generalizability claims.

**Questions:**

None

---

### Official Review · Reviewer_vJaV · 2025-10-28

**Soundness:** 2
**Presentation:** 2
**Contribution:** 2
**Rating:** 2
**Confidence:** 4

**Summary:**

This paper proposes BdSL-SPOTER, a pose-based Transformer for Bengali Sign Language (BdSL) recognition, with three adaptations: (1) signing-center normalization with a cultural scaling factor α, (2) motion-aware attention biasing that upweights low-motion “hold” frames, and (3) a prototype regularizer based on community-validated embedding clusters. On BdSLW60 the authors report signer-independent 5-fold CV with Top-1 = 94.2% ± 1.8%, plus ablations and a paired t-test versus SPOTER.

**Strengths:**

* Competitive results and ablations. Reported Top-1, Top-5, and Macro-F1 are high relative to listed baselines, and each cultural component contributes.
* Method description is clear enough to reproduce. The components are specified with equations and objectives, and the training pipeline is described at a level that enables re-implementation.

**Weaknesses:**

1. Causal support for “cultural” choices is limited. The α choice is motivated by a compactness gap relative to an ASL reference, but comparability across acquisition conditions is not established. A sensitivity study or a learnable α would strengthen the claim.
2. Low-motion attention bias relies on a strong linguistic assumption. Holds can also reflect hesitation or tracking noise. The paper does not show robustness of κ, γ, ε across sign speeds or styles.
3. Ethics statement is minimal. It notes anonymized consent but does not provide institutional approval details, recruitment and compensation, or data governance specifics.
4. Baseline coverage is inadequate. The paper does not compare against several widely used state-of-the-art sign language recognition baselines, both pose-based and RGB-based, under the same signer-disjoint protocol. Without these comparisons, the strength of the reported gains is unclear.
5. Lack of cross-corpus evidence. Results are shown only on BdSL. External validity remains uncertain without at least one non-BdSL benchmark or a controlled cross-dataset transfer study.

**Questions:**

* **About the “undervalued BdSL” claim.** If the core issue is resource scarcity, would a larger and more diverse BdSL dataset plus standard models close the gap? In other words, **what linguistic properties of BdSL make it fundamentally different from ASL or BSL in a way that still requires a distinct task framing when a dedicated dataset is available**? The introduction hints at compact signing space and longer holds, but a controlled cross-lingual study is not shown.
* **α selection and robustness.** Why α = 0.85, and how sensitive are results to camera distance, signer body size, and resolution? Consider a learnable or data-driven α.
* **Attention bias parameters.** Provide ranges and ablations for κ, γ, ε. Do fast signs or weak non-manual cues suffer under this bias?
* **Signer-disjoint folds.** Will you release exact fold files to enable leakage audits and exact replication? The appendix currently omits them.

**Details Of Ethics Concerns:**

* **Institutional approval for human-related research.** You report anonymized informed consent but do not state whether you obtained approval from an Institutional Review Board or a Human Research Ethics Committee. **Please confirm whether you received formal approval to conduct human-subject research, and provide the committee name, approval ID, and date.** This is critical given the use of community participants and interpreters.
* **Recruitment and compensation.** Describe recruitment channels, inclusion and exclusion criteria, compensation, withdrawal rights, and conflict of interest management for the 3 interpreters and 12 community participants.
* **Data governance.** Specify license ter

---

### Official Review · Reviewer_v8Uo · 2025-10-28

**Soundness:** 3
**Presentation:** 2
**Contribution:** 2
**Rating:** 2
**Confidence:** 5

**Summary:**

This paper introduces BdSL-SPOTER, a transformer-based model for Bengali Sign Language (BdSL) recognition, with cultural adaptation. The proposed model incorporates signing-space normalization, motion-aware attention biasing, and a community-derived prototype regularizer, specifically tailored to address the unique challenges of BdSL recognition.

**Strengths:**

1. Cultural Adaptation: The model addresses cultural differences in BdSL through novel techniques, including cultural regularization and motion-aware attention biasing. These methods help the model better understand and adapt to BdSL’s unique signing conventions, making it culturally sensitive.

2. Dataset and Experimental Design: The paper uses the BdSLW60 dataset, which contains 9,307 videos, ensuring a diverse and representative dataset. The rigorous signer-independent 5-fold cross-validation approach ensures the reliability of the experimental results and provides a strong evaluation framework.

3. High Recognition Accuracy: The model achieves an impressive 94.2% Top-1 accuracy on the BdSLW60 dataset. This demonstrates significant improvement over previous methods, showcasing the model’s effectiveness in BdSL recognition tasks.

**Weaknesses:**

1. Single Dataset Limitation: Although the BdSLW60 dataset is a valuable resource, the model’s evaluation is based on a single dataset. The lack of cross-dataset validation limits the generalizability of the model. Future work could benefit from expanding the model’s evaluation to multiple datasets to assess its broader applicability.

2. Limited Signer Diversity: While the BdSLW60 dataset includes 18 signers, the diversity of the signers (in terms of age, gender, and regional variations) is still limited. Greater diversity in the signer population could improve the robustness and generalizability of the model.

3. Cultural Adaptation Explanation: The paper introduces cultural regularization and motion-aware attention biasing as key components, but the explanation of how these mechanisms adapt to cultural differences is somewhat limited. A deeper exploration of how these adaptations work and how they reduce the impact of cultural variance would strengthen the paper.

4. Limitations and Future Work: The authors acknowledge the limitations of the current work, such as the use of a single dataset and limited signer diversity. However, the paper does not provide specific solutions or directions for addressing these issues in future work. More concrete strategies for cross-dataset validation, enhancing signer diversity, and improving cross-linguistic generalization could be outlined.

**Questions:**

See above **Weaknesses**.

---

### Meta-Review · Area_Chair_2MZg · 2025-12-03

**Summary:**

The authors study transformer-based approaches for Bengali Sign Language recognition and observe excellent empirical performance on one data set. Although the reviewers appreciate the reproducibility and performance of the approach, the paper has several shortcomings. For example, the approach could be better motivated, coverage of related work is insufficient and the empirical comparison lacks state-of-the-art baselines.

**Reviewer Concerns:**

There is no rebuttal.

**Reviewer Scores:**

There is no rebuttal and I don't think the paper would have received significantly higher scores if there was a discussion. There are too many crucial issues (e.g., missing baselines).

---

### Decision · Program_Chairs · 2026-01-26

Reject